# Cloning and Functional Characterization of a Flavonoid Transport-Related MATE Gene in Asiatic Hybrid Lilies (*Lilium* spp.)

**DOI:** 10.3390/genes11040418

**Published:** 2020-04-12

**Authors:** Hua Xu, Panpan Yang, Yuwei Cao, Yuchao Tang, Guoren He, Leifeng Xu, Jun Ming

**Affiliations:** 1Institute of Vegetables and Flowers, Chinese Academy of Agricultural Sciences, Beijing 100081, China; ahxuhua@163.com (H.X.); yangpanpan1988@126.com (P.Y.); 13067739520@163.com (Y.C.); tangyuchao100@126.com (Y.T.); hgr0222@sina.com (G.H.); 2College of Life Science, Gannan Normal University, Ganzhou 341000, China

**Keywords:** *Lilium*, anthocyanins, transportation, MATE, *LhDTX35*

## Abstract

Previous studies have suggested that multidrug and toxic compound extrusion (MATE) proteins might be involved in flavonoid transportation. However, whether MATE proteins are involved in anthocyanin accumulation in *Lilium* is unclear. Here, a flavonoid transport-related MATE candidate gene, *LhDTX35*, was cloned from the Asiatic hybrid lily cultivar ‘Tiny Padhye’ by rapid amplification of 5’ and 3’ cDNA ends (RACE) and found to encode 507 amino acids. BLASTx results indicated that *LhDTX35* showed high homology to the *DTX35* genes of other species. Bioinformatics analysis predicted that the protein encoded by *LhDTX35* possessed 12 typical transmembrane segments and had functional domains typical of the MATE-like superfamily. Phylogenetic analysis grouped *LhDTX35* in the same clade as the *DTX35* of other species. Notably, the expression pattern of *LhDTX35* was positively correlated with floral anthocyanin accumulation in ‘Tiny Padhye’. A subcellular localization assay showed that the protein encoded by *LhDTX35* was plasmalemma localized but not nuclear, indicating that the *LhDTX35* gene may function as a carrier protein to transport anthocyanins in *Lilium*. Functional complementation of the *Arabidopsis*
*DTX35* gene demonstrated that *LhDTX35* could restore silique-infertility and the anthocyaninless phenotype of an *Arabidopsis*
*DTX35* mutant. These results indicated that *LhDTX35* might be involved in anthocyanin accumulation in *Lilium.*

## 1. Introduction

Lilies (*Lilium* spp.) have high commercial and ornamental value due to their various floral colors and coloration patterns. Anthocyanins are one of the major pigments affecting the coloration of the floral tepals in *Lilium* spp. [1,2,3,4]. Anthocyanins are synthesized in the cytosol but their transfer to the vacuole is necessary for plant tissues to exhibit brilliant colors [5,6]. The mechanism of anthocyanin transfer is still unclear, although anthocyanin biosynthesis at the molecular level has been studied extensively leading to the discovery of enzyme-coding structural and regulatory genes related to anthocyanin biosynthesis [7,8]. To date, glutathione S-transferases (GSTs), multidrug and toxic compound extrusion (MATE) proteins, and the ATP-binding cassette (ABC) transporter have been proposed to be involved in anthocyanin transportation in many species [9,10,11].

MATE transporters perform various transport functions in plants, including the transportation of anthocyanin [12]. A MATE transporter related to flavonoid accumulation was isolated for the first time during screening of a *TT12 Arabidopsis* mutant with altered seed coloration [13]. Subsequently, orthologs of *AtTT12* were obtained from many other species, such as upland cotton, blueberry, *Medicago truncatula*, apple, and grapevine, and orthologs of the *TT12* gene were verified to be related to flavonoid transportation [6,14,15,16,17]. To date, the enzyme-coding structural genes and regulatory genes of anthocyanin biosynthesis in *Lilium* have been characterized thoroughly [2,4,18,19,20]; however, whether MATE transport is involved in anthocyanin transportation in *Lilium* remains unclear.

In this study, based on our RNA-seq data published previously [21], we isolated a flavonoid transport-related MATE gene, *LhDTX35*, from *Lilium* ‘Tiny Padhye’. Then, expression profiling, identification, and functional analysis of this gene in anthocyanin transportation were performed using ‘Tiny Padhye’ as a subject. This study demonstrates that a gene encoding MATE is critical for anthocyanin accumulation in *Lilium*, and promotes an understanding of the mechanism underlying MATE involvement in anthocyanin transportation in *Lilium*.

## 2. Materials and Methods

### 2.1. Plant Material

The Asiatic lily cultivar ‘Tiny Padhye’ was grown in a greenhouse (25 °C, relative humidity 60%–70%, 16 h/8 h light/dark cycles) at the Chinese Academy of Agricultural Sciences (Beijing, China). The upper parts and bases of the inner tepals were collected at four different developmental stages. The tepal developmental stages were defined as described in Xu LF et al. [21]: Stage 1 (S1: no anthocyanin pigment is visible in tepals), stage 2 (S2: anthocyanin pigment becomes visible on tepals), stage 3 (S3: the day before anthesis, lower halves of tepals are fully pigmented), and stage 4 (S4: the first day after anthesis) (Figure 1). Samples were obtained from 15 flowers and pooled together as one biological sample; three independent biological replicates were collected for each stage.

*Nicotiana tabacum* was also grown in a greenhouse, as described above for environmental conditions. Seedlings at the four-true-leaf stage (6-week-old) were used for the subcellular localization experiment.

### 2.2. Isolation of a MATE-Like Gene

Six primers (LhMATE-YZ-F1, LhMATE-YZ-R1, LhMATE-3’RACE-GSP, LhMATE-3’RACE-nest, LhMATE-5’RACE, and LhMATE-5’RACE-nest) were designed based on unigenes encoding MATE-like proteins from our previous RNA-seq experiment (Table 1). First-strand cDNA was cloned using RNA isolated from the colored parts of the tepal at stage 2 as a template, according to the RACE kit manufacturer’s instructions (Takara, Dalian China). To obtain the full-length MATE cDNA, the 5’-end and 3’-end sequences of the identified gene were identified using stage 2 total RNA from the basal (colored) tepal regions and a RACE kit, according to the recommendations of the manufacturer (Takara, Dalian, China). Five independent RACE cDNA clones were sequenced by Sangon Biotech (Shanghai, China). Based on the 5’-RACE and 3’-RACE sequencing results, the primers LhDTX35-F and LhDTX35-R (Table 1) were designed to amplify the entire gene region containing the open reading frame (ORF) from the first-strand cDNA.

### 2.3. Sequence Bioinformatics Analysis

A BLASTx search of MATE-like gene ORF sequences was conducted to analyze sequence similarity. Protein conserved domain prediction was performed by NCBI conserved domain online software [22]. The protein transmembrane domains were analyzed by TMHMM Server V.2.0 [23].

### 2.4. Phylogenetic Analysis

The amino acid sequences of genes encoding MATEs from different plants were used for phylogenetic analysis. Sequence alignment was performed using CLUSTAL X. Based on this alignment, a Neighbor-Joining (NJ) tree was constructed using MEGA (version 7.0). Bootstrap values were calculated using 1000 replicate analyses.

### 2.5. Gene Expression Analysis

Quantitative real-time polymerase chain reaction (qRT-PCR) was performed using SYBR Premix Ex Taq (Takara, Dalian, China). The primers used to amplify the MATE-like gene segment are shown in Table 1. The reaction parameters were as follows: (1) 95 °C for 1 min; (2) 40 cycles of 95 °C for 20 s, 60 °C for 10 s, and 72 °C for 25 s; and (3) a melt curve program (65 °C to 95 °C with an increment in temperature of 0.5 °C every 0.05 s). The signal was monitored using a CFX96 real-time system (Bio-Rad, CA, USA). The average Cq value was calculated from three biological and three technical replicates. To normalize the differences in the amounts of mRNA from other genes, the amount of *Lhactin* mRNA was determined for each sample and the relative expression level of *LhDTX35* was analyzed using the 2^−△△CT^ method [24]. The error bars represented the ±SEs from three independent experiments. The data were analyzed by ANOVA using SAS software.

### 2.6. Subcellular Localization Analysis

Using the transient expression vector pCAMBIA2300-35s-GFP, the subcellular localization of the *LhDTX35* gene was analyzed. To construct the expression vector 35S::*LhDTX35*-GFP, primers containing the XbaI and BamHI sites were designed to amplify the *LhDTX35* gene ORF domain using the primer LhDTX35-DW-F/R (Table 1). The PCR products were separated on a 1% agarose gel and then purified. The empty vector pCAMBIA2300-35s-GFP was digested with the restriction enzymes XbaI and BamHI according to the manufacturer’s instructions (Thermo Scientific FastDigest, America). Subsequently, the purified PCR product and cleaved empty vector were fused using T4 DNA Ligase (TransGen, China). Positive clones were selected and confirmed by sequencing and then transformed into *Escherichia coli* DH5α cells (TransGen, China). The sequencing primers (DW-YZ-F/R) are shown in Table 1. Finally, the plasmids pCAMBIA2300-GFP and pCAMBIA2300-GFP-*LhDTX35* were each transformed into *Agrobacterium tumefaciens* (LBA4404) using the freeze-thaw method, as previously reported [25].

When *N. tabacum* had developed four true leaves, the underside of the leaves was infiltrated with *Agrobacterium* inocula using a 1 mL needleless syringe. The inoculated plants were grown in a phytotron for 24 h in the dark (25 °C, relative humidity 60%–70%). Then, the plants were grown under 16 h/8 h light/dark cycles for 5 days. Subsequently, the LhDTX35-GFP fusion protein in the *N. tabacum* epidermis cells was microscopically detected using a Leica confocal laser scanning microscope. The samples were illuminated with an argon ion laser using 488 nm light for GFP and a green HeNe laser using 543 nm light for chlorophyll autofluorescence.

### 2.7. Complementation Analysis

The 35S::*LhDTX35* vector was constructed by replacing the GUS gene in the pCAMBIA3301 vector with the coding sequence of *LhDTX35*. The *LhDTX35* ORF was amplified by PCR with primers LhDTX35-noci-F and LhDTX35-Bahm-R (Table 1). The *LhDTX35* ORF fragments were cloned into the linearized vector pCAMBIA3301 and digested with the NocII and HindII restriction enzymes using a one-step seamless cloning kit (Transgen, Beijing, China). The binary vectors were introduced into *A. tumefaciens* GV3101 and subsequently used in *Arabidopsis ttDTX35* mutant transformation with the floral dip method [26]. Seeds of the *Arabidopsis LhDTX35* mutant, transgenic lines, and wild-type *Arabidopsis* were germinated and grown on Murashige & Skoog Basic Medium 1/2 Macro (1/2 MS) medium. Transgenic plants were screened on 1/2 MS medium plates that contained 50 mg/L kanamycin.

The expression of *LhDTX35* was analyzed by RT-PCR with primers LhDTX35-YZ-F/R, and *Arabidopsis Actin* gene was control with primers Actin-F/R (Table 1).

## 3. Results

### 3.1. Cloning and Sequence Analysis of the LhDTX35 Gene

The full-length cDNA sequence of *LhMATE*-like was obtained by RACE and deposited in GenBank with the accession number MT001433. This sequence contained 1948 bp and encoded 507 amino acids, as predicted by the NCBI ORF Finder tool (Figure 2). BLASTx results showed high homology to the *DTX35* genes of other species, such as *Dendrobium catenatum*, *Elaeis guineensis*, *Phoenix dactylifera*, and *Oryza sativa* Japonica Group (ca. 71%–76.36% identity at the amino acid level). Therefore, the *LhMATE*-like cDNA isolated in this study was named *LhDTX35*. Conserved domain prediction revealed that *LhDTX35* encoded functional domains typical of the MATE-like superfamily between 258 bp and 1562 bp (Figure 3). Transmembrane domain analysis predicted that the protein encoded by *LhDTX35* was a membrane protein with twelve typical transmembrane domains (Figure 4). A phylogenetic analysis revealed that *LhDTX35* was closely related to the DTX35 protein in *Dendrobium catenatum*, sharing 76.36% amino acid identity (Figure 5). In conclusion, the isolated *LhDTX35* gene was predicted to encode a membrane protein belonging to the MATE-like superfamily.

### 3.2. Expression Analysis of LhDTX35

Anthocyanin accumulated in only the basal tepal regions of ‘Tiny Padhye’. We previously showed that anthocyanin content increased gradually during tepal development (Figure 6) [21]. Anthocyanin accumulation was the highest in tepals at stage 3 (Figure 6) [21]. To exploit whether the *LhDTX35* gene was involved in anthocyanin accumulation in *Lilium*, qRT-PCR was performed using the upper and basal tepal regions at stages 1–4. These results showed that *LhDTX35* gene expression was positively correlated with anthocyanin accumulation, which occurred in the bases of the tepals (Figure 7).

### 3.3. Subcellular Localization of LhDTX35

A subcellular localization experiment was performed to further identify whether the protein encoded by the *LhDTX35* gene was a membrane protein. The results showed that GFP fluorescence was distributed in the cytoplasm of *N. tabacum* leaf epidermal cells treated with the empty GFP vector (Figure 8a,b). In contrast, GFP fluorescence was located in the membrane of *N. tabacum* leaf epidermis cells treated with the *LhDTX35* and GFP fusion protein vector (Figure 8c,d). These results indicated that the *LhDTX35* gene encoded a membrane protein.

### 3.4. Functional Analysis of the LhDTX35 Gene in the Arabidopsis DTX35 Gene Mutant

An *Arabidopsis* genetic mutant lacking *DTX35* was selected to investigate the functionality of the *LhDTX35* gene. The coding sequence of the *LhDTX35* gene was transferred into the *Arabidopsis DTX35* mutant under the control of the cauliflower mosaic virus (CaMV) 35S promoter. The hypocotyls of wild-type plants and transgenic lines expressing *LhDTX35* were red. In contrast, the hypocotyls of the *Arabidopsis DTX35* mutant were green, except for the junctions between the cotyledons (Figure 9B). The silique fertility of the *Arabidopsis DTX35* mutant was low but normal fertility was restored in the complementation experiment (Figure 9C). These results showed that the function of the *LhDTX35* gene was similar to that of the *DTX35* gene in *Arabidopsis*. Furthermore, the *LhDTX35* gene may be involved in anthocyanin accumulation.

## 4. Discussion

*Lilium* has high ornamental value due to its wide variety of coloration patterns. Anthocyanins are important pigments responsible for *Lilium* flower pigmentation ranging from pink to purple. Therefore, the characterization of anthocyanins in *Lilium* is of considerable ongoing interest. In *Lilium*, the steps of anthocyanin biosynthesis and regulation are well established; however, the anthocyanin transportation mechanism remains unclear. In recent years, several transport proteins have been reported to be involved in anthocyanin transportation [9,10,11]. Among them, MATE is an important protein that transports anthocyanins. To date, whether the MATE protein is involved in anthocyanin transportation in *Lilium* is unknown.

In the present study, a flavonoid transport-related MATE candidate gene, *LhDTX35*, was cloned successfully based on our previous transcriptome data. Phylogenetic analysis revealed that the *LhDTX35* gene clustered with *DTX35* genes from *Dendrobium catenatum, Phoenix dactylifera,* and *Oryza brachyantha* (Figure 5). *LhDTX35* is closely related to the *DTX35* gene in *Dendrobium catenatum*, sharing 76.36% amino acid identity. Bioinformatics analysis revealed that the *LhDTX35* gene contained sequences encoding the conserved domains typical of the MATE superfamily (Figure 3), suggesting that the protein encoded by *LhDTX35* belonged to the MATE family. In addition, MATEs are important transport proteins that possess 12 typical transmembrane domains, and transmembrane structure prediction showed that *LhDTX35* also has 12 typical transmembrane domains (Figure 4). These results indicated that *LhDTX35* is a MATE protein in *Lilium*.

The expression profiles of anthocyanin biosynthetic genes and regulatory genes during anthocyanin accumulation at stages 1–4 were previously analyzed in ‘Tiny Padhye’, and the results showed that anthocyanin accumulation in tepals increased gradually at stages 1–4, reaching its highest level at stage 3 [21]. In this study, we revealed that the expression level of *LhDTX35* was consistent with anthocyanin accumulation at stages 1–4 (Figure 7). Weak expression of *LhDTX35* was detected in the tepals at stage 1, when the anthocyanin content was lowest. In conjunction with the increasing anthocyanin content in the tepals, the transcript levels of *LhDTX35* were gradually upregulated at stages 1–4, and both anthocyanin content and *LhDTX35* gene transcript levels peaked at stage 3 (Figure 6 and Figure 7). All the results indicated that *LhDTX35* expression was positively correlated with anthocyanin accumulation in *Lilium*, suggesting that the *LhDTX35* gene may be involved in the transport of anthocyanin in *Lilium*.

All the plant MATE proteins characterized to date have been localized to membranes [27]. To investigate whether *LhDTX35* is involved in anthocyanin accumulation, its subcellular protein localization was examined. The results showed that LhDTX35 was localized to the membrane but not the cytoplasm, further suggesting that the *LhDTX35* gene encoded a protein with the traits of MATE transport proteins. These results were also consistent with the results of the transmembrane structure prediction, that is, the protein encoded by *LhDTX35* was predicted to have 12 typical transmembrane domains (Figure 4).

Complementation experiment is an effective method to analyze gene function because genetic transformation is a challenge for *Lilium*. Complementation has been used for this purpose in many species [9,11,15]. In this study, the silique-infertility phenotype of the *DTX35* mutant was recovered when *LhDTX35* was overexpressed in the *DTX35* mutant of *Arabidopsis*, suggesting that the *LhDTX35* gene has a similar function to that of *DTX35* in *Arabidopsis*. The *DTX35* gene is also called FFT (flower flavonoid transporter) [28]. The anthocyanin content of the *FFT* mutant was slightly less than that of the WT (Wile Type) in seedlings at one week of age [28]. Interestingly, the present study showed that the anthocyaninless phenotype of the *DTX35* mutant was rescued by the complementation experiment (Figure 9B). These results suggested that the *LhDTX35* gene was required for anthocyanin transportation in *Lilium*. MATE genes are part of a gene superfamily, including 58 MATEs in *Arabidopsis* and 53 MATEs in rice [29]. In the present study, only one MATE gene was cloned successfully, and whether any other MATE genes are involved in anthocyanin transportation in *Lilium* should be further studied.

## 5. Conclusions

In conclusion, we first identified a flavonoid transport-related MATE candidate gene based on transcriptome data in *Lilium* ‘Tiny Padhye’ and found that this gene shared high similarity with published *DTX35* genes. Bioinformatics and phylogenetic analysis revealed that *LhDTX35* belongs to the MATE gene family. Expression profiling showed that the expression of *LhDTX35* was positively correlated with anthocyanin accumulation in the tepals of ‘Tiny Padhye’. Subcellular localization results showed that the *LhDTX35*-encoded protein was localized to the membrane, suggesting that the protein encoded by the *LhDTX35* gene possessed the traits of a MATE transporter. Silique-infertility and anthocyaninless hypocotyl phenotypes were recovered after the complementation experiment, indicating that *LhDTX35* has a similar function to that of the *DTX35* gene in *Arabidopsis* and is involved in anthocyanin transportation in *Lilium*. Our results provide insight into the mechanism of anthocyanin transportation in *Lilium*.

## Figures and Tables

**Figure 1 genes-11-00418-f001:**
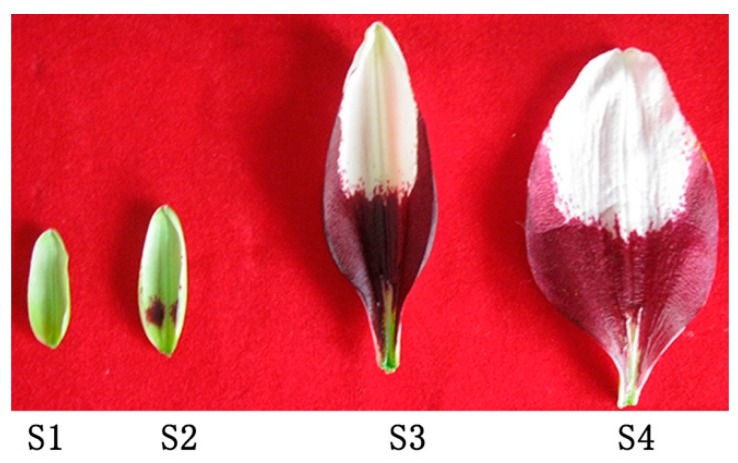
Tepals of the Asiatic lily cultivar ‘Tiny Padhye’ at four different developmental stages.

**Figure 2 genes-11-00418-f002:**
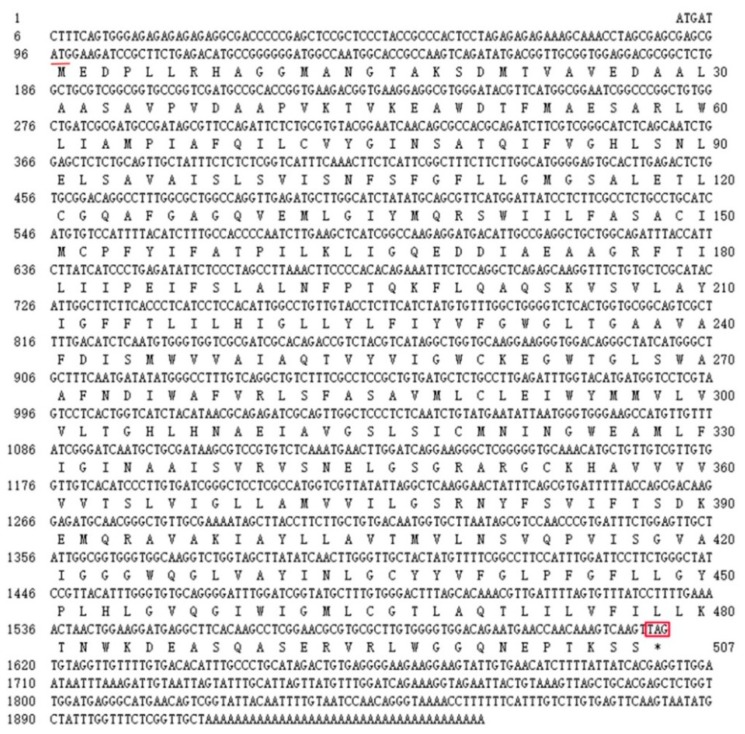
Nucleotide and deduced amino acid sequences of the *LhDTX35* gene. The initiation codon is underlined, while the stop codon is labeled with an asterisk and box.

**Figure 3 genes-11-00418-f003:**
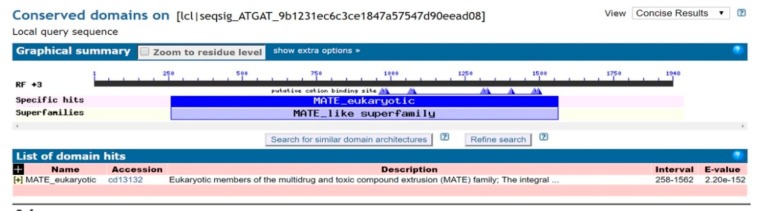
Conserved domain prediction for the protein encoded by *LhDTX35.*

**Figure 4 genes-11-00418-f004:**
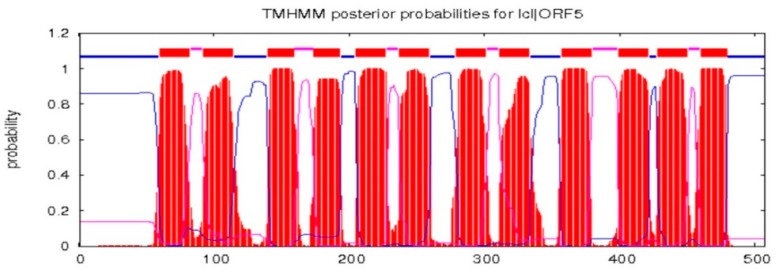
Transmembrane prediction for the protein encoded by *LhDTX35.*

**Figure 5 genes-11-00418-f005:**
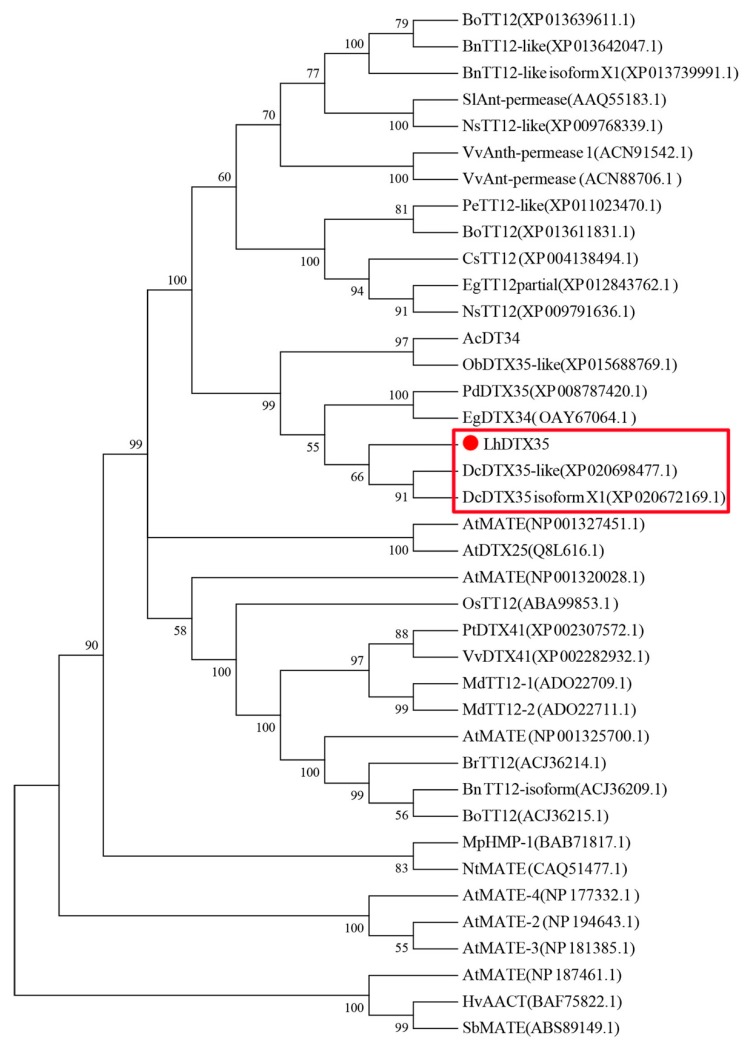
Phylogenetic tree of the predicted amino acid sequences of the *Lilium* multidrug and toxic compound extrusion (MATE) protein and MATE proteins from other plants. The phylogenetic tree was constructed using MEGA 6.0 software and the NJ method. The red dot indicates the *Lilium* MATE-like protein identified in this study.

**Figure 6 genes-11-00418-f006:**
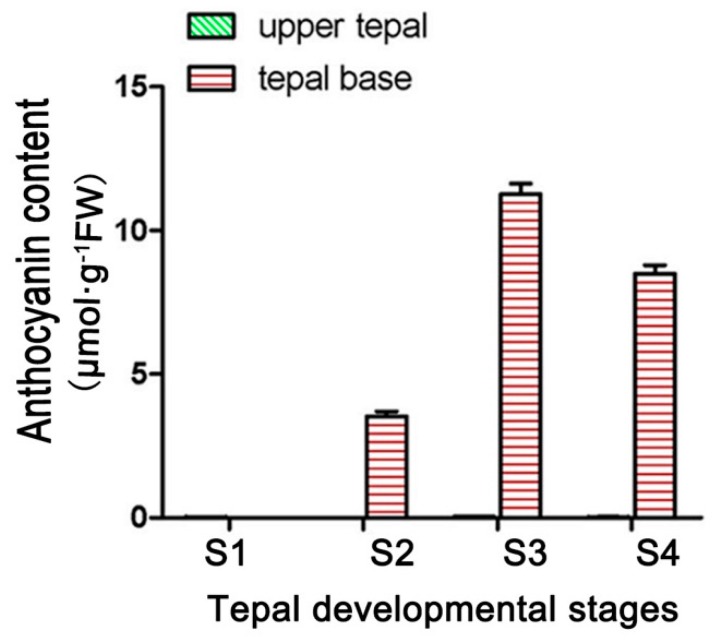
Changes in anthocyanin content at different developmental stages in *Lilium* [21].

**Figure 7 genes-11-00418-f007:**
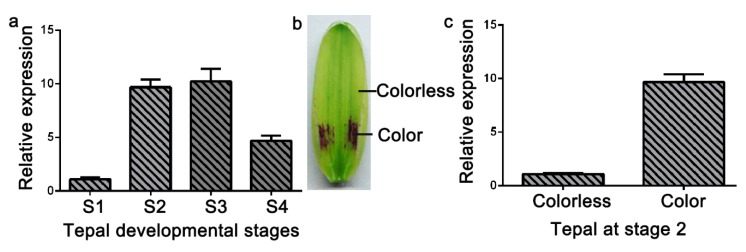
Spatiotemporal analysis of *LhDTX35* gene expression at different stages of tepal development in *Lilium* ‘Tiny Padhye’. (**a**) Expression profiles of *LhDTX35* at stages 1–4 in the basal tepal region in *Lilium* ‘Tiny Padhye’; (**b**) tepal at stage 2; (**c**) expression profiles of *LhDTX35* at stage 2 between the upper and basal tepal regions in *Lilium* ‘Tiny Padhye’.

**Figure 8 genes-11-00418-f008:**
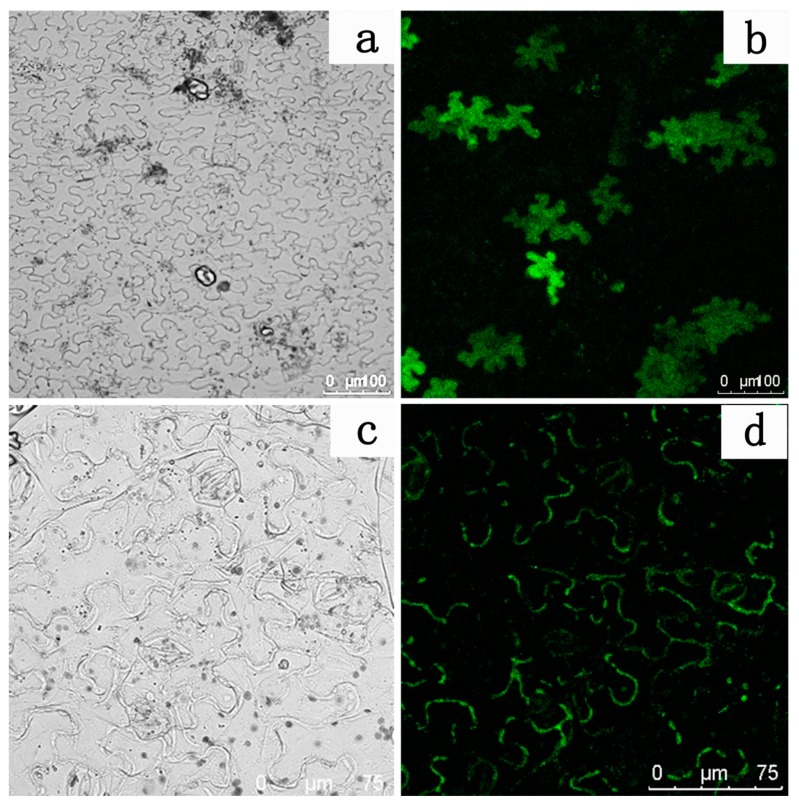
Subcellular localization of the *LhDTX35*-GFP fusion protein. (**a**,**c**) Bright field images; (**b**,**d**) GFP fluorescence images; (**a**,**b**) cells expressing the *LhDTX35*-GFP fusion protein; (**c**,**d**) cells expressing 35S::GFP as a control.

**Figure 9 genes-11-00418-f009:**
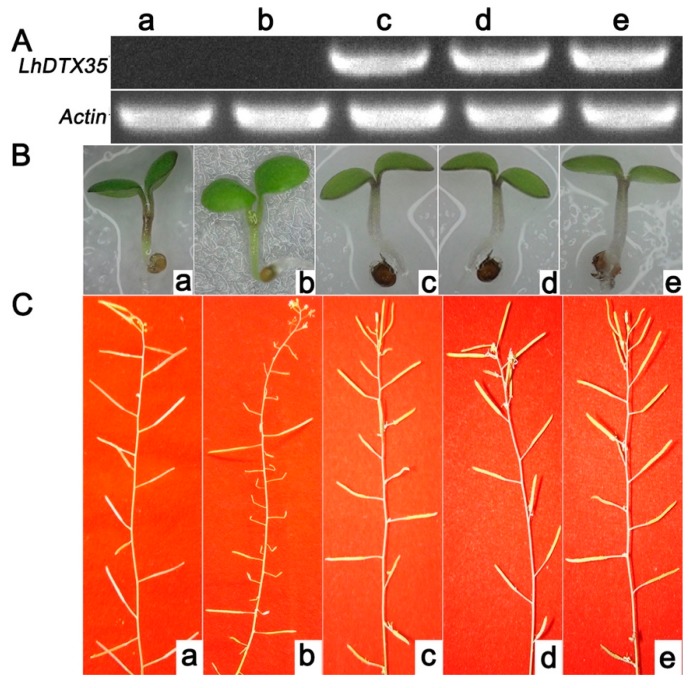
Functional complementation of the *Arabidopsis DTX35* mutant with the *LhDTX35* gene driven by the CaMV 35S promoter. (**A**) Expression of *LhDTX35* in wild-type (**a**), mutant (**b**), and transgenic *Arabidopsis* seedlings (**c**–**e**). (**B**) Anthocyanin accumulation in wild-type (**a**), mutant (**b**), and transgenic *Arabidopsis* seedlings (**c**–**e**). (**C**) Silique fertility of wild-type (**a**), mutant (**b**), and transgenic *Arabidopsis* seedlings (**c****–e**).

**Table 1 genes-11-00418-t001:** All primers used in this study.

Primers	Primer Sequence (5′-3′)
LhMATE-YZ-F1 LhMATE-YZ-R1	GTTGCGGTTATCACATCCCT CTTCTTCCCCTCACAGTCTA
LhDTX35-3’RACE-GSP LhDTX35-3’RACE-nest	CGGTGGGTGGCAAGGTCTGGTAGC GGGCTATCCGTTACATTTGGGTGTGC
LhDTX35-5’RACE, LhDTX35-5’RACE-nest	CCAAATCCCCTGCACACCCAAA TGGTTCATTCTGTCCACCCCAC
LhDTX35-F, LhDTX35-R	AGTGGGAGAGAGAGAGAGGCGA CTTCTTCCCCTCACAGTCTATGC
LhDTX35-YG-F LhDTX35-YG-R	CCAAGAGGATGACATTGCCGAG TGGAGGATGAGGGTGAAGAAGC
LhDTX35-DW-F LhDTX35-DW-R	CCCCCGGGATGGAAGATCCGCTTCTGAGAC GCTCTAGACACTAACTTGACTTTGTTGGTT
DW-YZ-F DW-YZ-R	CGGGCTGTTGCGAAAATA TGCCGTTCTTCTGCTTGTC
LhACT-YG-F LhACT-YG-R	GCATCACACCTTCTACAACG GAAGAGCATAACCCTCATAGA
Actin-F Actin-R	CGTGACCTTACTGATTACCT AGCGATACCTGAGAACATAG
LhDTX35-noci*-F* LhDTX35-Bahm-R	CGGGGGACTCTTGACCATGGAAGATCCGCTTCTGAGACAT GGAAATTCGAGCTGGTCACCTGTAATCTAACTTGACTTTGTTGGTT
LhDTX35-YZ-F LhDTX35-YZ-R	CGGGCTGTTGCGAAAATA TGCCGTTCTTCTGCTTGTC

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
