# Peer review of "Cloning and Functional Characterization of a Flavonoid Transport-Related MATE Gene in Asiatic Hybrid Lilies (Lilium spp.)"

_genes, 2020, doi:10.3390/genes11040418_

Round 1

Reviewer 1 Report

This study will enrich the MATE-based transport mechanisms of anthocyanin transportation in Lilium to develop new cultivars with high ornamental value.

Please address the following issues:

L43: Medicago truncatula (Italic)

L68: Indicate the age of seedlings at the four-true-leaf stage

L115: Agrobacterium tumefaciens delete “Agrobacterium strain”

L11: Leaf disk?

L118: Inoculate plants(Agrobacterial culture inoculated on the surface of leaves?)

L117-120: Not clear to me, please rewrite the method

L131: please indicate how T1 was selected? Like selection media used and the presence of transgene LhDTX35 was verified.

L156, 158: Figure 5 Lilium (Italic)

L160: Please explain why the previous anthocyanin data used. Though the authors used the same cultivar, the accumulation of anthocyanin may be affected by greenhouse conditions. Thus, the estimation of anthocyanin should be determined in the materials used for mRNA isolation (stage 2).

L181: Please improve the figure 8 quality

L188-190: Move to materials and methods

L191: Please determine the content of anthocyanin in transgenic and mutant Arabidopsis to support the LhDTX35 mediated anthocyanin accumulation.

Author Response

 Dear Editors and Reviewer 1,

Thank you for your letter and for the reviewers’ comments concerning our manuscript entitled “Cloning and Functional Characterization of a Flavonoid Transport-related MATE Gene in Asiatic Hybrid Lilies (Lilium spp.)” (Genes-764716). Those comments are all valuable and very helpful for revising and improving our paper, as well as the important guiding significance to our researches. We have studied the comments carefully and have made corrections which, we hope, can meet with your approval. Revised portion are marked in red in the paper. The main corrections in the paper and the responses to the reviewer’s comments are as the following: 

Responses to the reviewer’s comments: Reviewer 1:

Point 1: L43: Medicago truncatula (Italic)

Response 1: We have made correction in Line 43 according to the reviewer’s suggestion. The “Medicago truncatula” in Line 43 was corrected with Italic. The corrections were marked in red.

Point 2: L68: Indicate the age of seedlings at the four-true-leaf stage

Response 2: We have made correction in Line 68 according to the reviewer’s suggestion. The age of seedlings at the four-true-leaf stage has been added in Line 68.

Point 3: L115: Agrobacterium tumefaciens delete “Agrobacterium strain”

Response 3: We have made correction in Line 115 according to the reviewer’s suggestion. “Agrobacterium strain” has been replaced by Agrobacterium tumefaciens.

Point 4: L11: Leaf disk?

Response 4: We have rewritten the method. The contents were that “When N. tabacum had developed four true leaves, the underside of the leaves was infiltrated with Agrobacterium inocula using a 1-mL needleless syringe. The inoculated plants were grown in a phytotron for 24 h in the dark (25°C, relative humidity 60-70%). Then, the plants were grown under 16 h/8 h light/dark cycles for 5 days.”

Point 5: L118: Inoculate plants (Agrobacterial culture inoculated on the surface of leaves?)

Response 5: We have rewritten the method. The contents were that “When N. tabacum had developed four true leaves, the underside of the leaves was infiltrated with Agrobacterium inocula using a 1-mL needleless syringe. The inoculated plants were grown in a phytotron for 24 h in the dark (25°C, relative humidity 60-70%). Then, the plants were grown under 16 h/8 h light/dark cycles for 5 days.”

Point 6: L117-120: Not clear to me, please rewrite the method

Response 6: We have rewritten the method. The contents were that “When N. tabacum had developed four true leaves, the underside of the leaves was infiltrated with Agrobacterium inocula using a 1-mL needleless syringe. The inoculated plants were grown in a phytotron for 24 h in the dark (25°C, relative humidity 60-70%). Then, the plants were grown under 16 h/8 h light/dark cycles for 5 days.”

Point 7: L131: please indicate how T1 was selected? Like selection media used and the presence of transgene LhDTX35 was verified.

Response 7: We have added the method of selecting the Transgenic plants and verifying the presence of transgene LhDTX35. The contents were that “Transgenic plants were screened on 1/2 MS medium plates that contained 50 mg/L kanamycin. The expression of LhDTX35 was analyzed by RT-PCR with primers LhDTX35-YZ-F/R and Arabidopsis Actin gene was control with primers Actin-F/R (Table 1.).”

Point 8: L156, 158: Figure 5 Lilium (Italic)

Response 8: We have corrected “Lilium” in Line58 with Italic.

Point 9: L160: Please explain why the previous anthocyanin data used. Though the authors used the same cultivar, the accumulation of anthocyanin may be affected by greenhouse conditions. Thus, the estimation of anthocyanin should be determined in the materials used for mRNA isolation (stage 2).

Response 9: As to issue why the previous anthocyanin data used, the reasons are as follows: Firstly, we used the same cultivar; secondly, the cultivars were planted in the same greenhouse under the same conditions; Thirdly, the tepal developmental stages were defined as described in Lei Feng Xu et al(Xu L et al, 2017). Fourthly, results of anthocyanin content analysis at different tepal stage in our lab indicated very stable in ‘Tiny padhye’. Fifthly, Lei Feng Xu and Hua Xu worked in the same lab, and the previous anthocyanin analysis was done by both Lei Feng Xu and Hua Xu. Therefore, the previous anthocyanin data was used in the paper.

Point 10: L181: Please improve the figure 8 quality

Response to 10: I am very sorry we can’t replace figure 8 as figure 8 is the best picture we have taken. In the later research, we will improve our ability to take picture.

Point 11: L188-190: Move to materials and methods

Response to 11: We have made corrections in Line 188-190 according to the reviewer’s suggestion. The contents between line 188 to line 190 have been moved to line 130-line 131. 

Point 12: L191: Please determine the content of anthocyanin in transgenic and mutant Arabidopsis to support the LhDTX35 mediated anthocyanin accumulation.

Response to 12: As to issue of point 12, reasons are as follows: Previous results showed that in AtDTX35 mutant seedlings up to 1week of age, extracted anthocyanins were slightly reduced in level compared with the WT (Thompson, 2009), meanwhile, the colors of the hypocotyls in WT, AtDTX35 mutant and transgenic lines can be distinguish by phenotype. Therefore, here, we did not determine the content of anthocyanin again. 

We tried our best to improve the manuscript and made some changes in the manuscript. These changes will not influence the content and framework of the paper. And here we did not list the changes but marked in red in revised paper. We appreciate for Editors/Reviewers’ warm work earnestly, and hope that the correction will meet with your approval. Once again, thank you very much for your comments and suggestions.

Reviewer 2 Report

Xu et al. present a study on a MATE gene, LhDTX35, isolated from Lilium 'Tiny Padhye'. They cloned the gene sequence by using a unigene sequence from a previous RNA-seq project that allowed them to decipher the full gene sequence using 5' and 3' RACE. The authors conducted phylogenetic analysis of LhDTX35 compared to MATE proteins in other species and protein domain analyses to show that LhDTX35 is likely to be a MATE protein. qRT-PCR expression analysis showed correlation with anthocyanin production during tepal development. Transient transformation in tobacco leaves showed localization to the plasma membrane and overexpression in Arabidopsis without functional DTX35 complemented the mutant phenotype, suggesting that LhDTX35 has similar function to AtDTX35. Taken together, the results from Xu et al. indicates that they have isolated a MATE gene from Lilium 'Tiny Padhye' that is involved in transporting anthocyanin.

I think this was a well-written and well-organized paper. I think that the experiments done and the results obtained fully support their conclusion that LhDTX35 is likely to be an important MATE protein involved in anothocyanin transport in Lilium 'Tiny Padhye'. I also agree with the authors' acknowledgement (line 250) that the other MATE genes in the genome will have to be investigated in future experiments in order to understand which MATE genes besides LhTX35 are involved in anthocyanin transport.

Author Response

 Dear Editors and Reviewer 2,

       Thank you for your letter and for the reviewers’ comments concerning our manuscript entitled “Cloning and Functional Characterization of a Flavonoid Transport-related MATE Gene in Asiatic Hybrid Lilies (Lilium spp.)” (Genes-764716). Those comments are all valuable and very encouraging, as well as the important guiding significance to our future researches. We appreciate for Editors/Reviewers’ hard work earnestly. Once again, thank you very much for your comments.

                                                                                               Mingjun

                                                                                              2020-4-6